# Chimeric Antigen Receptor T Cell Therapy Targeting Epithelial Cell Adhesion Molecule in Gastric Cancer: Mechanisms of Tumor Resistance

**DOI:** 10.3390/cancers15235552

**Published:** 2023-11-23

**Authors:** Yanping Yang, Raymond Louie, Janusz Puc, Yogindra Vedvyas, Yago Alcaina, Irene M. Min, Matt Britz, Fabio Luciani, Moonsoo M. Jin

**Affiliations:** 1Department of Radiology, Houston Methodist Research Institute, Houston, TX 77030, USAimin@houstonmethodist.org (I.M.M.); 2Molecular Imaging Innovations Institute, Department of Radiology, Weill Cornell Medicine, New York, NY 10065, USA; yalcaina@ratiotx.com; 3School of Computer Science and Engineering, University of New South Wales (UNSW), Sydney, NSW 2052, Australia; r.louie@unsw.edu.au; 4AffyImmune Therapeutics, Inc., Natick, MA 01760, USA; 5Department of Surgery, Weill Cornell Medicine, New York, NY 10065, USA; 6School of Medical Sciences and Kirby Institute for Infection and Immunity, University of New South Wales (UNSW), Sydney, NSW 2052, Australia

**Keywords:** CAR T cell therapy, EpCAM, single-cell multiomics, CAR T cell imaging

## Abstract

**Simple Summary:**

The specific mechanisms by which tumors acquire resistance to chimeric antigen receptor (CAR) T cell therapy are not completely understood. The aim of this study was to elucidate the complex interactions between tumor cells and CAR T cells targeting epithelial cell adhesion molecule (EpCAM) in a xenograft model of gastric cancer. Using whole-body CAR T cell imaging and single-cell multiomic analyses, we noticed that within resistant tumors, CAR T cells exhibited a tendency to proliferate, but they were largely dysfunctional, losing their ability to fight cancer effectively. Specifically, most CD8 T cells became exhausted within tumors, while CD4 T cells transformed into regulatory T cells that can dampen the immune response. Additionally, the resistant tumor cells had specific gene changes that could promote cancer growth and make the disease more challenging to cure. This research provides valuable information for understanding how tumors resist CAR T cell therapy and may guide future developments in cancer treatment.

**Abstract:**

Epithelial cell adhesion molecule (EpCAM) is a tumor-associated antigen that is frequently overexpressed in various carcinomas. We have developed chimeric antigen receptor (CAR) T cells specifically targeting EpCAM for the treatment of gastric cancer. This study sought to unravel the precise mechanisms by which tumors evade immune surveillance and develop resistance to CAR T cell therapy. Through a combination of whole-body CAR T cell imaging and single-cell multiomic analyses, we uncovered intricate interactions between tumors and tumor-infiltrating lymphocytes (TILs). In a gastric cancer model, tumor-infiltrating CD8 T cells exhibited both cytotoxic and exhausted phenotypes, while CD4 T cells were mainly regulatory T cells. A T cell receptor (TCR) clonal analysis provided evidence of CAR T cell proliferation and clonal expansion within resistant tumors, which was substantiated by whole-body CAR T cell imaging. Furthermore, single-cell transcriptomics showed that tumor cells in mice with refractory or relapsing outcomes were enriched for genes involved in major histocompatibility complex (MHC) and antigen presentation pathways, interferon-*γ* and interferon-*α* responses, mitochondrial activities, and a set of genes (e.g., *CD74*, *IDO1*, *IFI27*) linked to tumor progression and unfavorable disease prognoses. This research highlights an approach that combines imaging and multiomic methodologies to concurrently characterize the evolution of tumors and the differentiation of CAR T cells.

## 1. Introduction

Dysfunctional T cells with exhaustion or tolerance phenotypes are the main cause for the inability of the immune system to reject many cancers. Persistent antigenic stimulation either under chronic viral infection or in response to tumors leads to the downregulation of T cell receptor (TCR) expression and ultimately the loss of effector functions [1,2]. Exhausted T cells are characterized by defective cytokine production (e.g., IL-2, IFN-*γ*, and TNF-*α*), impaired proliferation, distinct transcriptional and epigenetic gene signatures, and high expression of inhibitory receptors (e.g., PD-1, LAG-3, and TIM-3) [1,3]. Recent studies leveraging single-cell gene expression and surface marker analyses have identified a subset of dysfunctional (PD1, LAG3, and TIM3) yet proliferating (Ki-67) CD8 tumor-infiltrating lymphocytes (TILs) in human non-small cell lung cancer (NSCLC) and melanoma tumors, which is distinct from previously described exhausted T cells with a loss of proliferative capacity [4,5]. Similar observations have been reported in murine melanoma tumor models [6]. It is not clear whether T cells engineered to express chimeric antigen receptors (CARs) follow a similar fate as tumor-infiltrating endogenous T cells, due to the design feature of CAR T cells incorporating both TCR activation and co-stimulation properties.

T cell exhaustion induced by immune checkpoint activation, the acquisition of a regulatory T (T_Reg_) cell phenotype by CD4 T cells, and the loss of cell surface CAR expression have been identified as the major mechanisms for inadequate CAR T cell potency against tumors [7,8,9,10]. The incorporation of genetic circuits to enable the expression of cytokines such as IL-12 and IL-15 [11,12,13] and dominant negative forms of PD-1 and TGF-*β* receptors [14,15,16], the amelioration of the loss of surface CAR [17], and regulatable CAR activity [18] have shown promise in rescuing CAR T cells from exhaustion and sustaining antitumor activity. Therefore, detailed characterization of the complex tumor–T cell interplay and a deeper understanding of mechanisms leading to T cell dysfunction and tumor resistance could enable improved strategies for harnessing T cell immunity to fight cancer.

In recent years, multiomic analyses, including genomics, transcriptomics, and TCR repertoire profiling, have played a pivotal role in identifying key genes involved in T cell cytotoxicity and dysfunction [19,20], with the characterization of CAR T cell functional states that are associated with cell products, clinical responses, and toxicities [9,21,22], and profiling of different TCR clonotypes and their kinetics [23]. Compared with extensive studies conducted for CAR T cells against hematological malignancies, multiomic studies focusing on CAR T cells against solid tumors, which face distinct challenges in the immunosuppressive tumor microenvironment (TME), have been limited primarily to in vitro and immunocompetent animal models [24,25,26,27].

We previously developed CAR T cells targeting epithelial cellular adhesion molecule (EpCAM) for the treatment of gastric cancer [28]. While we found that adding intercellular adhesion molecule 1 (ICAM-1) targeting can enhance EpCAM CAR T cell activity, resistance and tumor recurrence remained significant challenges. In this current study, our objective was to delve into the mechanisms behind tumor resistance to CAR T cells. Our research specifically concentrated on elucidating the direct interactions between human xenograft tumor cells and CAR T cells within a gastric cancer xenograft model. We combined a whole-body CAR T cell imaging technique with single-cell multiomic analyses, encompassing single-cell RNA sequencing, surface marker detection, and TCR V(D)J sequencing. The incorporation of the spatiotemporal kinetics of tumor and CAR T cells into omic profiling holds significant promise for enhancing our understanding of cell–cell interactions that influence the treatment response and toxicity. Our longitudinal positron emission tomography (PET) imaging of CAR T cells and bioluminescence imaging of tumor cells revealed continuous interactions between the two. Within the resistant tumors, we observed dynamic tumor cell death and proliferation, while CAR T cells exhibited continuous expansion. The single-cell multiomics analyses suggested that resistant tumors had induced specific signaling pathways to promote their survival, and tumor-infiltrating CAR T cells had acquired tolerance and exhaustion.

## 2. Materials and Methods

### 2.1. Cell Lines

HEK293T cells were obtained from the American Type Culture Collection (ATCC) and cultured in Dulbecco’s Modified Eagle’s Medium (DMEM, Corning, NY, USA) supplemented with 10% heat-inactivated fetal bovine serum (FBS, GeminiBio, West Sacramento, CA, USA). The human gastric cancer cell line MKN-45 was purchased from DSMZ and cultured in RPMI-1640 (Corning) supplemented with 10% FBS. GFP–firefly luciferase (FLuc)-expressing MKN-45 cells were generated by lentiviral transduction (Biosettia, San Diego, CA, USA) for bioluminescence imaging in the animal model. All cells were maintained in a humidified incubator at 37 °C with 5% CO_2_ and routinely tested for mycoplasma contamination (MycoAlert^TM^ mycoplasma detection kit, Lonza, Lexington, MA, USA).

### 2.2. Lentiviral Vector Construction

CAR antigen-binding domains were derived from the monoclonal antibody UBS54 for EpCAM and affinity-tuned (F292A) I domain of human lymphocyte function-associated antigen 1 (LFA-1) for ICAM-1 [29,30,31]. The EpCAM-specific single CAR (UBS54-CD28-CD3ζ) and EpCAM×ICAM-1 Bi-dual CAR (UBS54-CD28-CD3ζ–T2A–F292A-4-1BB-CD3ζ) were generated as previously described [28]. The Tan-dual CAR (F292A-USB54-CD28-CD3ζ) was generated by joining the F292A I domain with UBS54 scFv using a (G_4_S)_2_ linker, followed by CD8 hinge, the transmembrane and cytoplasmic domains of CD28, and the cytoplasmic domain of CD3ζ. A c-Myc tag was introduced at the N-terminus of each CAR for the detection of CAR expression. The PET reporter gene SSTR2 was incorporated after the CAR following a “self-cleaving” ribosome-skipping porcine teschovirus-12 A (P2A) sequence. CAR constructs were cloned by GenScript into a lentiviral plasmid backbone (VectorBuilder Inc., Chicago, IL, USA) under the regulation of a human elongation factor 1α (EF-1α) promoter.

### 2.3. Transduction of Human Primary T Cells

Lentiviral vectors were packaged by VectorBuilder and stored at −80 °C until use. Healthy donor human leukopaks were purchased from Biological Specialty Corporation. CAR T cells were manufactured as previously described [28]. Briefly, CD4/CD8-enriched T cells were activated with human T-expander CD3/CD28 Dynabeads (Gibco, Waltham, MA, USA) at a bead-to-cell ratio of 1:1 and cultured at 1 × 10^6^ cells/mL in complete T cell growth medium: TexMACS medium (Miltenyi Biotec, Gaithersburg, MD, USA) containing 5% human AB serum (Sigma, Livonia, MI, USA), IL7 (12.5 ng/mL, Miltenyi Biotec), and IL15 (12.5 ng/mL, Miltenyi Biotec). The T cells were transduced twice with lentivirus (1:25 dilution) at 24 and 48 h after activation. T cell cultures were inspected daily and maintained at 1–3 × 10^6^ cells/mL. On day 10, CAR T and non-transduced control T cells were cryopreserved in a 1:2 mixture of T cell growth medium and CryoStor CS10 (Biolife Solutions, Bothell, WA, USA).

### 2.4. Flow Cytometry

T cell product samples were washed with PBS containing 0.5% BSA and blocked with mouse IgG (200 µg/mL, Sigma-Aldrich, St. Louis, MO, USA, cat. no. I8765), and then stained with c-Myc antibody (FITC, 1:50 dilution, Miltenyi Biotec, clone SH126E7.1.3) at 4 °C in the dark for 15 min. The cells were washed twice before analysis using a Gallios flow cytometer (Beckman Coulter, Brea, CA, USA) and the Gallios cytometry list mode data acquisition and analysis software (Beckman Coulter). Flow cytometry data were analyzed using FlowJo v.10.8.1 (Tree Star, Inc., Ashland, OR, USA). Dead cells were excluded based on forward and side-scatter gating. Gating for CAR-positive cells was based on non-transduced T (NT) cells with the same staining.

### 2.5. Mouse Experiments

All animal experiments were performed according to protocols approved by the Weill Cornell Medicine Institutional Animal Care and Use Committee (ICAUC# 2012-0063, 3-year renewal approved in 2019). Six- to eight-week-old male NOD.Cg-Prkdc^scid^ Il2rg^tm1Wjl^/SzJ (NSG) mice were purchased from the Jackson Laboratory, and housed in the Animal Core Facility at Weill Cornell Medicine (New York, NY, USA). The mice were maintained under pathogen-free conditions. NSG mice were subcutaneously injected with 1 × 10^6^ FLuc-expressing MKN-45 cells suspended in 100 μL of RPMI-1640/Matrigel (1:1) into the upper left flank. Five days later, the mice were randomized on the basis of bioluminescent imaging or tumor size measurements to ensure tumor establishment before treatment. Subsequently, the mice were divided into 5 groups, which received 10 × 10^6^ NT, UBS54, Bi-dual, or Tan-dual CAR T cells, or no treatment at all (*n* = 4 individual mice per group). Tumor growth was monitored weekly using an IVIS Spectrum in vivo imaging system (PerkinElmer, Waltham, MA, USA). Bioluminescence images were acquired 15 min after intraperitoneal injection of 200 μL of 15 mg/mL D-luciferin (Gold Biotechnology, Olivette, MO, USA). Tumor volume (V) was measured weekly with a caliper and calculated as V = [length × (width)^2^]/2. PET-CT imaging was performed to track CAR T cell biodistribution using a micro-PET-CT scanner (Inveon, Siemens, Washington, DC, USA) 2 h after intravenous injection of the ^18^F-NOTA-OCT tracer (1,4,7-Triazaclononane-1,4,7-triacetic acid-octreotide). ^18^F-NOTA-OCT was prepared as previously described [31]. The health condition of the mice was monitored on a daily basis by the veterinary personnel, independent of the investigators and studies. Euthanasia was applied when a humane endpoint had been reached (e.g., >25% body weight loss, signs of illness or distress including ruffled fur, difficulty eating, or abnormal posture). The investigators were not blinded when performing imaging and monitoring tumor responses. All animals were included in the data analysis.

### 2.6. Tumor Dissociation and Single Cell Isolation

One mouse from each treatment cohort was sacrificed on day 21 and fresh tumor tissues were collected and minced into small pieces using a scalpel and digested at 37 °C for 2 h using the human tumor dissociation kit (Miltenyi Biotec, 130-095-929) following the manufacturer’s instructions. After digestion, the tumors were mechanically disrupted and filtered through a 70 μm cell strainer (Corning). The dissociated cells were subsequently washed with cold 1× PBS containing 0.5% bovine serum albumin (BSA) and 2 mM EDTA. Erythrocytes were removed using 5 mL of ACK lysing buffer (Gibco). The cells were then resuspended in cold 1× PBS containing 0.5% BSA and 2 mM EDTA and incubated with 20 μL of human CD45 (TIL) MicroBeads (Miltenyi Biotec, 130-118-780) per 10⁷ cells for 15 min at 4 °C. Magnetically labeled cells were sorted using a MACS LS column (Miltenyi Biotec, 130-042-401) in the magnetic field of a MACS Separator. The flow-through unlabeled cells were collected as CD45-depleted tumor samples. After removing the column from the separator, the magnetically retained CD45^+^ cells were eluted as the positively selected TILs.

### 2.7. Single-Cell Sequencing Procedure

Single-cell RNA sequencing (scRNA-seq) was performed using a Chromium Next GEM 5′ Single Cell V(D)J Reagent Kit v1.1 from 10x Genomics according to the manufacturer’s protocol. In brief, MKN-45 tumor (tumor cell line and xenograft cells) and T cells (CAR T cell products and TILs) were counted and labeled with barcoded antibodies (TotalSeq-C purchased from BioLegend, San Diego, CA, USA) to identify different cell types by expressed cell surface markers (CD3, CD4, CD8, CD25, CD127, ICAM-1, and EpCAM). The labeled cells were resuspended to 1000 cells per µL, and captured using Single Cell Chip G on a 10x Chromium Controller (10x Genomics) to generate gel bead-in emulsions (GEMs). Reverse transcription was performed using a C1000 Touch Thermal Cycler (Bio-Rad, Hercules, CA, USA). Barcoded complementary DNA (cDNA) was recovered through post-GEM-RT cleanup steps and PCR amplification. The recovered cDNA was amplified and used to construct 5′ whole-transcriptome (5′ GEX), V(D)J, and cell surface protein libraries. The quality and quantity of cDNA and each library was assessed using High Sensitivity DNA reagents and an Agilent 2100 Bioanalyzer (Agilent, Santa Clara, CA, USA). The 5′ GEX and V(D)J libraries were indexed using a Chromium i7 Sample Index Kit, and the cell surface protein libraries were indexed using a Chromium i7 Sample Index Plate N, Set A kit. 5′ GEX libraries were pooled at a 10x ratio of the V(D)J and of the cell surface protein libraries and sequenced on a NovaSeq 6000 System and an S4-300 cycle lane (150PE) with v1.5 sequencing chemistry.

The following antibodies were used for scRNA-seq: TotalSeq-C0251 anti-human Hashtag 1 (BioLegend, Cat. no. 394661), TotalSeq-C0252 anti-human Hashtag 2, (BioLegend, Cat. no. 394663), TotalSeq-C0253 anti-human Hashtag 3 (BioLegend, Cat. no. 394665), TotalSeq-C0254 anti-human Hashtag 4 (BioLegend, Cat. no. 394667), TotalSeq-C0255 anti-human Hashtag 5 (BioLegend, Cat. no. 394669), TotalSeq™-C0256 anti-human Hashtag 6 (BioLegend, Cat. no. 394671), TotalSeq™-C0257 anti-human Hashtag 7 (BioLegend, Cat. no. 394673), TotalSeq™-C0258 anti-human Hashtag 8 (BioLegend, Cat. no. 394675), TotalSeq-C0049 anti-human CD3 (BioLegend, Cat. no. 344849), TotalSeq-C0045 anti-human CD4 (BioLegend, Cat. no. 344651), TotalSeq-C0046 anti-human CD8 (BioLegend, Cat. no. 344753), TotalSeq-C0085 anti-human CD25 (BioLegend, Cat. no. 302649), and TotalSeq-C0390 anti-human CD127 (BioLegend, Cat. no. 351356).

### 2.8. Pre-Processing of Single-Cell RNA-Seq, Protein, and TCR Data

Gene expression and TCR (10x 5′ V(D)J scRNA-seq), protein (TotalSeq), and demultiplexing hashtag matrices were produced using the Cell Ranger workflow (10x Genomics, v3.0.1). We utilized a modified GRCh38 reference genome which was appended with a sequence that corresponded to the transcript product of the CAR T cell transgene. Demultiplexing of the hash-tagged samples was first performed using Seurat’s HTODemux function. The samples were then quality controlled to remove cells with a low mitochondrial content, low/high number of unique genes, and total counts. Gene expression was normalized using scran with default parameters, while protein expression was normalized with centered-log-ratio (CLR) normalization. In the tumor samples, contaminating lymphocyte cells were removed by filtering out *CD3D*+*CD3E+* cells. Similarly, in the lymphocyte samples, contaminating epithelial cells were removed by filtering out *EPCAM*+ cells.

### 2.9. Integration, Clustering, and Identification of Subsets

Integration, dimensionality reduction, and clustering were performed using the Seurat R package [32]. Gene expression matrices was first integrated using the FindIntegrationAnchors and IntegrateData functions. The top 3000 highly variable genes were extracted using the FindVariableFeatures function. Principal component analysis was performed using the RunPCA function with the option npcs = 30, followed by neighbor and cluster calculations using the FindNeighbours and FindClusters functions, respectively. We explored resolution parameters in the range of 0.01 to 0.6 to determine the marker genes which adequately explained the observed clusters.

### 2.10. Differential Expression Analysis

Two differential expression analyses were performed depending on whether the analysis was performed within the same sample to find heterogenous clusters, or between different samples to account for inter-sample differences. In the first approach, for each sample, DGE (intra-sample) analyses were performed using the FindAllMarkers function with the “MAST” option [33]. The clusters used in this analysis were formed from the integrated data described in the previous section. Differentially expressed genes that appeared in the majority of samples were designated marker genes. This allowed the identification of markers which truly represented the clusters across most samples, in contrast to markers which were only differentially expressed in one sample. UMAP was then used to visualize the clustering results, using nearest neighbor = 30 and min_dist = 0.01. The dotplots were generated using ggplot2, with the averages obtained over all samples. In the second approach, DGE (inter-sample) analyses were performed using the edgeR wrapper run_de function in the Libra R library with the likelihood- ratio (LRT) and pseudobulk option [34]. The PAGA plots were generated by running pca (scanpy:pca), finding the nearest neighbours (scanpy:neighbours) and running PAGA (scanpy:paga) using default parameters on the integrated gene expression matrix [35]. Slingshot was applied to the UMAP formed from the integrated gene expression matrices using the getLineages and getCurves functions [36], and manually assigning an initial root node. This resulted in pseudotime values, which were used to generate the smoothed gene expression and Aucell vs scaled pseudotime curves, where the smoothed expressions were calculated using the geom_smooth R function with default parameters.

### 2.11. Gene Set Analysis

Gene set enrichment analysis (GSEA) was performed using the fgsea function in the fgsea R package [37] with default parameters, based on the DGE results ranked by the fold-change. Pathways and gene sets were downloaded from the MSigDB [38] in combination with curated T cell specific gene sets (Appendix A). Average NES values were obtained by averaging over the NES values in each sample.

### 2.12. Statistical Analysis

Statistical significance was determined by two-way ANOVA with Tukey’s multiple comparison test using Prism 9 (GraphPad Inc., San Diego, CA, USA). *p* values < 0.05 were considered statistically significant.

## 3. Results

### 3.1. CAR T Cell Imaging Reveals Penetration and Expansion of CAR T Cells in Tumor

The current understanding of tumor resistance to CAR T cell therapy implicates limited infiltration into and inadequate potency within tumors. To gain a deeper understanding of the resistance mechanisms, we used a gastric cancer model with MKN-45 xenografts that frequently exhibits tumor resistance or recurrence following CAR T cell treatment [28]. We performed a longitudinal PET-CT scan to monitor CAR T cell expansion and harvested tumors for single-cell analyses of both the tumor cells and tumor-infiltrating T cells (Figure 1A). Three CAR constructs were used in this study: (1) an EpCAM-specific single CAR (UBS54-CD28-CD3ζ) [28]; (2) a bicistronic dual CAR (labeled as Bi-dual) expressing two separate EpCAM and ICAM-1 CARs (UBS54-CD28-CD3ζ–T2A–F292A-4-1BB-CD3ζ) [28]; (3) a tandem dual CAR (labeled as Tan-dual) that joins an EpCAM-binding UBS54 scFv and ICAM-1-binding F292A I domain in the ectodomain to make a single CAR (F292A-USB54-CD28-CD3ζ) (Figure 1B). All CAR T cells were engineered to co-express somatostatin receptor 2 (SSTR2) to enable CAR T cell imaging in vivo using a SSTR2-specific PET tracer, ^18^F-NOTA-Octreotide [39]. EpCAM single, Bi-dual, and Tan-dual CAR T cells produced significant tumor responses against subcutaneous MKN-45, ranging from partial to complete with some animals displaying fluctuations in the tumor burden (Figure 1C–F). Compared with CAR T-treated cohorts, animals treated with non-transduced T (NT) cells showed a lack of tumor response, displaying a progression pattern similar to mice that received no T cell treatment (No T). In mice treated with CAR T cells, PET-CT imaging revealed the homing and expansion of CAR T cells in tumors, whereas mice receiving NT or no T cells did not exhibit a detectable uptake of ^18^F-NOTA-Octreotide into tumors (Figure 1D). The specific localization of CAR T cells and the fluctuation in the tumor burden suggest a specific and continuous interaction between CAR T cells and tumors.

### 3.2. Single-Cell Analysis of Tumor Cells Revealed a Remodeling towards External Stimuli following CAR T Cell Infusion

We next sought to understand the mechanism underlying the resistance of tumors despite the presence and expansion of tumor-infiltrating CAR T cells. To gain deeper insights into the crosstalk between CAR T cells and tumors, and the molecular and gene expression changes that ultimately lead to tumor resistance, we applied single-cell multiomic analyses to tumor and tumor-infiltrating lymphocytes (TILs) isolated from tumor xenografts to determine the surface markers, gene expression, and T cell clonal expansion (Figure 2A). To provide the baseline gene expression patterns, we included pre-xenograft MKN45 tumor cells and pre-infusion CAR T cell products (Figure 2A and Table 1). We included antibodies against T cells to identify T cell subsets (CD3, CD4, CD8) and activation (CD25, CD127). Single-cell 5′ gene expression and antibody sequencing combined with V(D)J enrichment were performed to relate the clonal expansion of T cells to gene and surface marker expression.

Clustering and dimensionality reduction of the single-cell gene and protein expression data of MKN-45 tumor cells identified six main clusters (Figure 2B). A correlation with cell cycle was observed, with clusters C2 and C3 predominantly in the G1 phase (Figure 2B). Each cluster contained cells from all samples. Clusters C2 and C4 exhibited a relatively even composition of cells from each sample (Figure 2C). In contrast, cluster C3 primarily consisted of cells from the pre-xenograft cell line and non-CAR T-treated xenografts (No T and NT) (Figure 2C,D). On the other hand, cluster C5 predominantly comprised tumor xenograft cells that had been treated with CAR T cells (Figure 2C,D). Cluster C0 was a dominant subset of pre-xenograft MKN-45 cells (~55%), and also accounted for approximately 30% of the MKN-45 cells isolated from xenografts (Figure 2C). This cluster displayed a signature of proliferating and activated cells, as evidenced by the prominent expression of genes such as Claudin-7 (*CLDN7*), which are associated with gastric cancer cell proliferation and invasion (Figure 2E and Appendix A) [40]. Cluster C1 was primarily represented by MKN-45 cells isolated from tumor xenografts (25–30% of total xenograft MKN-45 cells and less than 5% from pre-xenograft cells), and its defining gene set included keratin family genes (*KRT19*, *KRT8*), *CXCL1*, *CD9* (tetraspanin-29), *IFI27*, and IL-1 receptor (*IL1R2*), which are associated with inflammation (Figure 2E and Appendix A) [41]. Cluster C2 showed high expression of the transcription factors *EGR1* and *JUNB* and transcriptional regulator *NUPR1*, whereas cluster C4 contained cells expressing high levels of heat shock protein (HSP)-encoding genes, such as *HSPA1A* and *HSPA6* (HSP70 family), which are known to be involved in the pathology of gastric cancer (Figure 2E and Appendix A) [42]. Cluster C3 represented a small subset of cells and its representation was further reduced in tumors treated with CAR T cells (Figure 2C,D). These cells expressed *HSPA5*, *TAF1D*, and *TES*, and exhibited an overlapping signature with clusters C0 and C1 (Figure 2E). The most isolated cluster C5 displayed increased expression of *MKI67*, long non-coding RNA, *MALAT1*, and *NEAT*, as well as signatures associated with integrin cell surface interactions (Figure 2E and Appendix A).

To identify genes that exhibited differential expression in MKN-45 cells as a result of their interaction with CAR T cells, we performed a differential gene expression analysis of MKN-45 cells isolated from xenografts in the non-CAR T-treated control group (No T and NT) and CAR T cohorts (Figure 2F,G). With CAR T cell treatment, MKN-45 cells induced an upregulation of *IFI27* (interferon alpha-inducible protein 27), *TNFRSF14*, and *ISG15* (interferon-stimulating gene 15), a group of genes related to antigen processing and presentation (*CD74*, *HLA*-*B*, *HLA-DR*, and *TAP1*), as well as *CXCL1* and *CXCL2* genes that are associated with inflammasome activation and neutrophil recruitment (Figure 2G). *IFI27* has been implicated in cancer cell migration, invasion, and poor prognoses (Figure 2G) [43,44]. The Gene Set Enrichment Analysis (GSEA) performed on the differentially expressed genes revealed a group of pathways that were upregulated in MKN-45 cells from CAR T cohorts. These pathways included antigen binding and major histocompatibility complex (MHC), interferon gamma and alpha response, PD-1, and inflammatory response pathways (Figure 2F). The observed shift in gene expression patterns from non-CAR T-treated MKN-45 xenografts to the upregulated inflammatory, mitochondrial, and metabolic pathways from the CAR T cohort suggests a transition of tumor cells towards a survival-oriented response to the cytotoxic effects induced by tumor-infiltrating CAR T cells.

### 3.3. T Cell Analysis Revealed a Rapid Differentiation Process That Resulted in CD8 T Cell Exhaustion and CD4 T Cells Adopting a Regulatory Phenotype

We next examined the gene expression profiles of TILs, using pre-infusion NT, Bi-dual, and Tan-dual CAR T cell products as references. Through dimensionality reduction and a cluster analysis, we identified a total of nine distinct clusters, comprising CD4 T cell clusters T0 and T5 and CD8 T cell clusters T1 and T4 (Figure 3A). Clusters T2 and T3, as well as the smaller clusters T6, T7, and T8 comprised both CD4 and CD8 subsets. The majority of TILs were observed in clusters T0 (CD4) and T1 (CD8), whereas the pre-infusion NT and CAR T cell products were dominant in the clusters T2, T3, T4, and T5 (Figure 3B,C). From the TIL samples, approximately 60% of the CD4 T cells and 70% of the CD8 T cells were in T0 and T1, respectively. Effector cells were primarily found in clusters T7 and T8, while clusters T2 and T3 were dominated by pre-infusion NT and CAR T product cells and displayed an enrichment of genes associated with the cell cycle (Figure 3D). Furthermore, an analysis of cell cycle genes revealed a significant correlation between clusters and specific cell cycle states. The majority of CD4 and CD8 T cells were found in the G1 phase, while cluster T3 contained CD3 T cells in the S phase (Figure 3A). In contrast, cluster T2 exhibited an enrichment of genes associated with the G2M phase, indicating the rapid cycling of these cells. To validate our findings, we performed regression based on the cell cycle phase and confirmed the presence of proliferating cells among both CD4 and CD8 T cell populations without significant changes in the cluster distribution (Appendix A).

The analysis of surface markers and cluster genes revealed the molecular profile of pre-infusion T cell products and TIL samples in each cluster (Figure 3D and Appendix A). Cluster T0 predominantly comprised CD4 TILs exhibiting a T_Reg_ signature, characterized by the induction of *FOXP3* and exhaustion-associated genes such as *LAG3*, *PDCD1*, and *TIGIT* (Figure 3D). In contrast, cluster T5 was found to primarily comprise pre-infusion CD4 T cell products (Figure 3C), showing distinctive expression of *IL7R*, *TCF7*, *IRF1*, *JUN*, and *STAT1* (Figure 3D). The transition from T5 to T0 in TILs indicates a phenotypic change from effector to exhausted and regulatory T cells. CD8 TILs are the major constituents of cluster T1 and displayed a cytotoxic phenotype (*GZMB*, *PRF1*). However, these CD8 TILs also exhibited gene expression patterns indicative of T cell exhaustion (*LAG3*, *PDCD1*, *TIGIT*, and *SOX4*), bearing resemblance to the previously described cytotoxic and dysfunctional tumor-infiltrating T cells [5]. On the other hand, cluster T4 mainly comprised cytotoxic pre-infusion CD8 T cells (*GZMA*, *GZMB*), while lacking the expression of exhaustion-associated genes (*LAG3*, *PDCD1*, and *TIGIT*). Clusters T2 and T3 were characterized by an enrichment for genes related to the G2M and S phases, respectively (Figure 3A). These clusters expressed cell cycle genes such as *HMGB1* and *STMN1* (Figure 3D). Furthermore, cluster T2, which was enriched in cells in the G2M phase, expressed higher levels of genes associated with cell proliferation (*MKI67*). Lastly, the small clusters, T7 and T8, had higher expression levels of mitochondrial genes as well as of non-coding RNA transcripts (*MALAT1*, *NEAT1*) and the transcription factor *KLF6* (Figure 3D). A GSEA between clusters confirmed the gene expression profiles, for example, the enrichment of T regulatory function in cluster T0, and of exhaustion in T1, and notably enrichment in MYC targets in cluster T3 and a downregulation of the IL2 and STAT5 signaling pathways (Appendix A).

To identify the main features of TILs, we performed a differential gene expression analysis comparing TILs with the pre-infusion T cell products (Figure 3E). Our analysis revealed increased expression of the exhaustion markers *LAYN* and *PDCD1* in TILs. Additionally, *CXCL13*, which is known to be present in TILs upon concurrent T cell receptor (TCR) stimulation [45]; was also upregulated in our TIL samples. In contrast, pre-infusion T cells were enriched for *CISH* and *MAL*, which are known to negatively regulate TCR signaling (Figure 3E) [46,47]. The GSEA further validated our findings by identifying the enrichment of specific gene signatures in TILs and pre-infusion T cell products (Figure 3F). In TILs, we observed an enrichment in gene signatures associated with cytotoxicity, T cell exhaustion, NOTCH signaling, and CD4 T_Reg_ cells, indicating their functional characteristics within the tumor microenvironment. On the other hand, pre-infusion T cell products were enriched for gene signatures related to MYC targets, oxidative phosphorylation, and cell cycle signatures (e.g., G2M checkpoints).

### 3.4. Detection of CAR T Cell Transcript

To identify CAR T cells, we examined the expression of the transcripts that corresponded with the coding sequence of UBS54 scFv, which is present in all three CAR constructs. A small fraction (<2%) of NT cells showed CAR mRNA expression, which could be attributed to imprecise demultiplexing of the hash-tagged NT and CAR T cell samples (Figure 3G). In contrast, a substantial proportion of cells in both the pre-infusion CAR T cell products and TILs (up to 30%) were identified to be CAR-positive. These results are consistent with our flow cytometric data, which showed that 23% and 48% of pre-infusion Bi-dual and Tan-dual T cell products were CAR-positive, respectively (Figure 3H).

### 3.5. Analysis of TCR Sequences Revealed Clonally Expanded CAR T Cells in the Tumor

The presence of CAR T cells at tumor sites, coupled with their correlation with a reduced tumor burden, strongly suggests an active role of CAR T cells in tumor cell killing. To further support this observation, we examined clonal expansion within CAR T cells based on the identification of TCR*αβ* chains by V(D)J sequencing (Figure 4). Among the CD4 T cells, we detected a total of 31 and 183 expanded clones (a TCR sequence found in more than one cell) in the product and in TILs, respectively (Figure 4B,C). CD4 T cells exhibited prominent clonal expansion, predominantly within the T_Reg_ cluster T0 and cycling clusters T2 and T3 (Figure 4A,B). Notably, larger clones (size > 4) were observed only within CD4 TILs (Figure 4C). Similarly, clonal expansion was detected within CD8 T cells, with 95 and 60 expanded clones in the product and in TILs, respectively. Clonally expanded CD8 T cells were mostly within the exhausted cell cluster T1 and cycling clusters T2 and T3 (Figure 4D–F). Due to the limited sample size of this experiment, an exhaustive assessment of clonal expansion was not performed. A Shannon evenness analysis of the TCR*αβ* CDR3 amino acid sequences from CD4 T cells showed a reduced evenness within TILs, confirming an increased clonal expansion compared to the product T cells (Figure 4G). In contrast, the repertoires of CD8 T cells showed comparable evenness in the product and TILs, in line with a lower rate of clonal expansion (Figure 4H).

### 3.6. Trajectory Analysis Demonstrates a Differentiation of CAR T Cells towards Exhausted and T_Reg_ Cells

The dominance of clonally expanded T_Reg_ and exhausted CD8 T cells within CAR TILs, contrasting with the prevalence of cell cycling and proliferating features in the T cell product samples, suggests an underlying differentiation process associated with the tumor response. To quantify the molecular changes occurring in the CAR T cells and estimate the cell states through which T cells underwent differentiation, we conducted a pseudotime trajectory analysis using scRNA-seq data (Figure 5). We used a partition-based graph abstraction (PAGA) algorithm on the UMAP clusters to infer the putative connections between the inferred clusters. We first separated CD4 and CD8 T cells and focused on connected clusters, excluding the smaller isolated clusters T6, T7, and T8 (Figure 5A). For CD8 T cells, the trajectory analysis revealed a linear connection between the clusters, with cell cycle cluster T2 connected to T3 and the exhaustion cluster T1 (Figure 5B). Cluster T4, composed mainly of T cell products with a memory phenotype (Figure 3A,D,E), also exhibited a close connection with the exhausted T cells in cluster T1 (Figure 5B). This analysis suggested that the terminally differentiated cluster of exhausted T cells in T1 were connected to cells in the G2M (T2) and S (T3) cell cycle phases as well as with memory cells (T4). The analysis of CD4 T cells revealed a linear structure, similar to the CD8 compartment, with the memory cells in cluster T5, which is primarily comprised of T cell products (Figure 3A,C), connected to the T_Reg_ cluster T0 (Figure 5B). Additional connections between T_Reg_ cells and clusters T2 and T3 were also observed (Figure 5B). Overall, the PAGA analysis suggests a terminal differentiation process, wherein TILs develop a T_Reg_ and exhausted profile arising from both memory cells in the infusion products and cells undergoing rapid proliferation in the G2M and S phases of the cell cycle.

To assess the differentiation of CD8 T cells through the estimated cell states, we calculated the pseudotime values along the inferred connected clusters and obtained an estimate for the order with which T cells differentiate through the cell states. The inferred trajectory was obtained assuming a random cell in T2 as the root and terminating on the cluster T1. Along this trajectory, we calculated the average gene expression and gene modules and fitted these data using a locally estimated scatterplot smoothing (Loess) curve (Figure 5C). For CD8 T cells, we observed an increase in the mean expression of exhaustion-associated genes (Appendix A) as cells differentiated along the trajectory (Figure 5D). Conversely, gene signatures associated with metabolic activity (oxidative phosphorylation) and the IL-12 response exhibited a decline, while the IFN-*γ* response signature remained relatively stable. In comparison, the trajectory analysis of CD4 T cells reinforced the observations of an increase in *FOXP3* T_Reg_ gene expression in the T0 cluster. Additionally, there was an upregulation of TNF pathway gene signature scores and a decline in oxidative phosphorylation (Figure 5D). This analysis demonstrated a progressive transition along the inferred trajectory towards the exhausted and cytotoxic phenotypes for CD8 T cells, while CD4 T cells exhibited a transition towards the regulatory phenotypes (e.g., *FOXP3*).

## 4. Discussion

This study reports an approach to investigate the phenotypic evolution and communication between T cells and tumor cells that contribute to CAR T cell dysfunction and tumor resistance using single-cell multiomic and spatiotemporal CAR T cell imaging techniques. Our whole-body imaging of tumor growth and CAR T cell distribution revealed a continuous expansion of CAR T cells within growing tumors, suggesting the possibility of immune dysfunction or tolerance towards tumor cells. Through single-cell analyses of gene expression and surface markers in T cells, we found the gene signatures in T cells associated with CD4 T_Reg_ cells and cytotoxic yet exhausted CD8 T cells, while tumor cells showed an overexpression of genes related to metabolic activity, interferon response, and other pathways associated with tumor malignancy. The presence of dysfunctional yet proliferating T cells aligns with recent observations in TILs isolated from melanoma and NSCLC cells [4,5]. A study of CAR T cells targeting mesothelin in preclinical mouse models also identified proliferating and hypofunctional CAR T cells [7]. These tumor-infiltrating CAR T cells underwent rapid loss of functional activity, characterized by an upregulation of intrinsic T cell inhibitory enzymes and surface inhibitory receptors (PD1, LAG3, TIM3, and 2B4) [7]. By integrating single cell multiomics and spatiotemporal CAR T cell imaging techniques, our study provides further insights into the interaction between CAR T cells and tumors.

The longitudinal imaging of tumor and CAR T cells revealed persistent or dynamic expansion of CAR T cells and fluctuations in tumor growth. This finding suggests that CAR T cells were continuously engaged in tumor killing, although with limited efficacy, as we observed a rebound of tumor cells. Regulatory T cells were found to be the dominant cluster within CD4 TILs. Cytotoxic CD8 T cells were present within TILs; however, these cells mostly acquired an exhausted phenotype in proximity to the tumor. Additionally, TILs contained only a small proportion of cells that had a memory phenotype, which was prominent in pre-infusion T cell products. These results were consistent with the trajectory analysis of T cell differentiation, which revealed a transition from both memory cells and proliferating cells in the infusion products towards a cytotoxic state and then a dysfunctional state. Finally, clonal expansion in tumor-infiltrating CAR T cells further supported the terminal differentiation towards regulatory and exhausted subsets, and suggested that interactions between CAR T and tumor cells in the TME involved molecular mechanisms associated with both T cell activation and suppression. In our immunocompromised NSG model, CAR T exhaustion is likely driven primarily by chronic interactions between CAR T cells and persistent tumors. When antigens persist, such as in chronic infections and resistant cancers, T cells can enter a dysfunctional state [48]. Tonic signaling mediated by CARs can also accelerate T cell differentiation and exhaustion [49]. The design of the CAR construct, the promoter, and viral vector used to express the CAR should all impact the level of tonic signaling, thereby influencing T cell exhaustion [50,51]. Another contributing factor could be the inadequate vasculature within the rapidly growing xenograft tumors, leading to hypoxic conditions and contributing to the generation of exhausted CD8 T cells and T_Reg_ cells [52,53].

The use of single-cell multiomics from prior studies has shed light on some of the mechanisms that underlie CAR T cell–tumor interactions. For example, in immunocompetent mouse models, scRNA-seq has demonstrated that STING agonists and immunogenic chemotherapy can improve tumor control and enhance CAR T cell recruitment through the modulation of the TME [26,27]. Additionally, CAR T cell treatment of mouse syngeneic glioblastoma has been found to activate intratumoral myeloid cells and induce endogenous T cell memory responses [25]. In comparison, our study combined imaging and single-cell multiomics to provide a comprehensive quantitative measurement of the phenotype and molecular profiles of both CAR T cells and tumor cells in vivo, thus allowing us to identify the key features of the dynamical interactions between them. We uncovered heterogeneity in the distribution of cancer cells and molecular clues that are likely resulting from direct interactions with CAR T cells, for example, revealing immune modulatory responses such as increased antigen processing and presentation, TNF pathway activity, and HLA expression.

However, we also acknowledge several limitations of this study. Firstly, single-cell studies were conducted using tumor cells and TILs isolated from mice with sufficient tumor mass and TIL infiltration. Tumor-infiltrating CAR T cells in our study exhibited a phenotypic spectrum ranging from cytotoxic T cells at one end to exhausted T cells at the other, reflecting the partial suppression of tumor growth by CAR T cells. Our analysis did not include CAR T cells from mice with a complete tumor response, primarily due to technical challenges associated with collecting these T cells. After tumor elimination, CAR T cells often contract or migrate away from the tumor sites. Nevertheless, it is reasonable to speculate that CAR T cells from complete responders would exhibit a phenotypic spectrum skewed towards the cytotoxic end, featuring fewer exhausted and dysfunctional T cells. Secondly, our study used an immunocompromised NSG model, and as a result, it did not fully recapitulate the complexity of the tumor stroma and the interactions in the tumor stroma comprising T cells and other immune cells. Despite this limitation, our approach allowed us to focus on the investigation of cellular interactions between tumor cells and TILs without being complicated by other cellular components. Moreover, NSG tumor models have been widely utilized in the preclinical evaluation of CAR T cell therapy and offer greater clinical relevance compared to in vitro models [24,54]. Future work could apply this approach to a humanized mouse model with an intact immune system and assess CAR T cell responses over an extended period to provide insights into both the short-term and long-term dynamics. Another limitation is the relatively small number of TCR sequences analyzed for the TCR clonal analysis. This is likely to underestimate the clonal repertoire of responding CAR T cells.

## 5. Conclusions

This research introduces a tool to investigate the interactions between cancer and CAR T cells, and highlights the feasibility of integrating genomics and imaging techniques to investigate the evolutionary processes of both CAR T and cancer cells. It allows for the quantification of tumor size, and identification of phenotypic and transcriptional features of tumor and T cells, as well as the T cell clonal repertoire, enabling a comprehensive understanding of the molecular mechanisms driving a successful CAR T cell response.

## Figures and Tables

**Figure 1 cancers-15-05552-f001:**
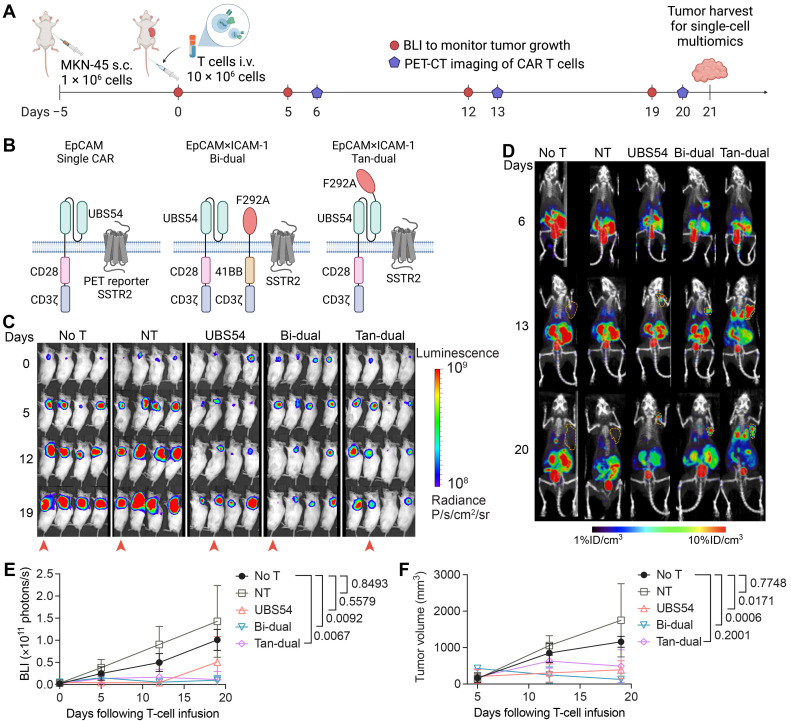
Tumor-infiltrating CAR T cells suppress progression of gastric cancer tumors. (**A**) Graphical overview of the experimental workflow. NSG mice were inoculated with 1 × 10^6^ MKN-45 cells subcutaneously and treated 5 days later with 10 × 10^6^ non-transduced T (NT) or CAR T cells by intravenous injection. Bioluminescence and PET-CT imaging were performed weekly to monitor tumor burden and CAR T cell expansion, respectively. Image created with BioRender.com. (**B**) Schematic representation of the EpCAM single CAR and EpCAM×ICAM-1 Bi-dual and Tan-dual CARs. Image created with BioRender.com. (**C**) Bioluminescent images of MKN-45-engrafted NSG mice. Animals subject to PET-CT imaging are indicated by red arrowheads. (**D**) Longitudinal PET-CT images of mice indicated by red arrowheads in C, showing accumulation of CAR T cells in tumors. Subcutaneous tumors are outlined using orange dashed lines. (**E**) Quantification of total body bioluminescence intensity. (**F**) Weekly tumor size measurements. Data in (**E**,**F**) represent mean ± SD of 4 mice per group. Statistical significance was determined by two-way ANOVA with Tukey’s multiple comparison test.

**Figure 2 cancers-15-05552-f002:**
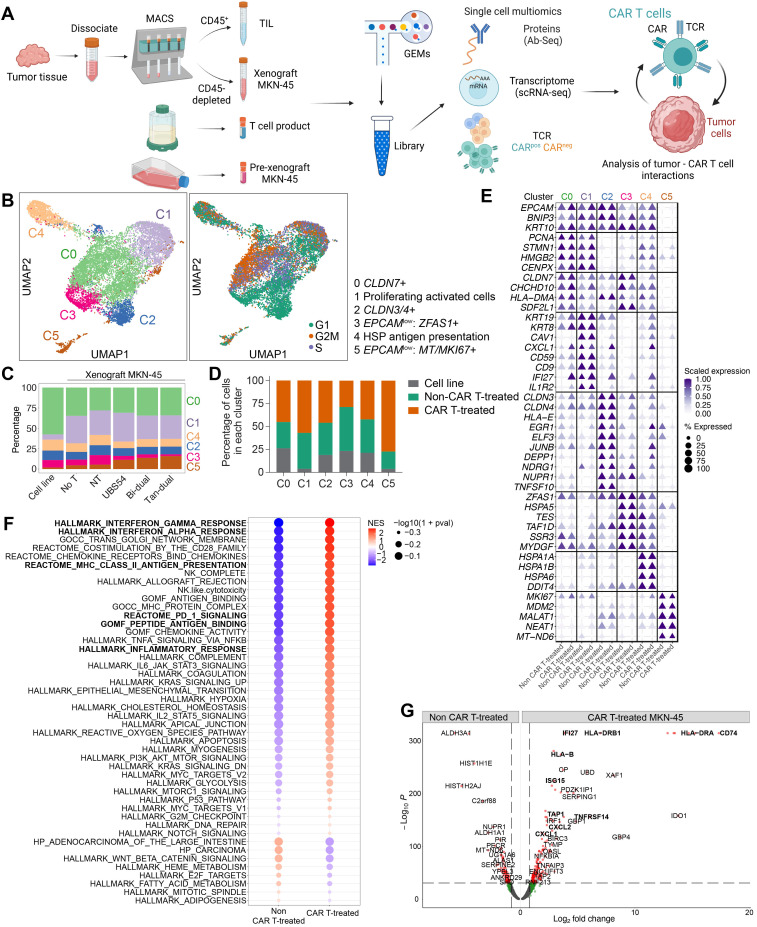
Single-cell multiomics of tumor cells are consistent with remodeling in response to external stimuli. (**A**) Overview of the experimental and sequencing workflow. Tumor samples were harvested and sorted into human CD45+ TILs and CD45– tumor cells for single-cell multiomic analyses (*n* = 1 mouse per treatment cohort). Pre-xenograft MKN45 tumor cells and pre-infusion CAR T cell products were included to obtain the baseline gene expression profiles. (**B**) Transcriptomics-based UMAP of MKN45 cells showing the presence of six main clusters. The plot on the right represents cell cycle phases projected onto the UMAP. (**C**) Proportions of cells within each sample origin. Colors represent clusters identified from the UMAP analysis. (**D**) Proportions of cells within each cluster. The non-CAR T-treated group comprised MKN-45 xenograft cells without treatment (No T) and those treated with non-transduced T (NT) cells. The CAR T-treated group consisted of MKN-45 xenografts treated with UBS54, Bi-dual, or Tan-dual CAR T cells. (**E**) Dot plot representing cluster genes measured as average gene expression normalized by origin of sample. (**F**) GSEA of the differential gene expression data. (**G**) Volcano plot of top differentially expressed genes when comparing CAR T-treated versus non-CAR T-treated MKN-45 cells.

**Figure 3 cancers-15-05552-f003:**
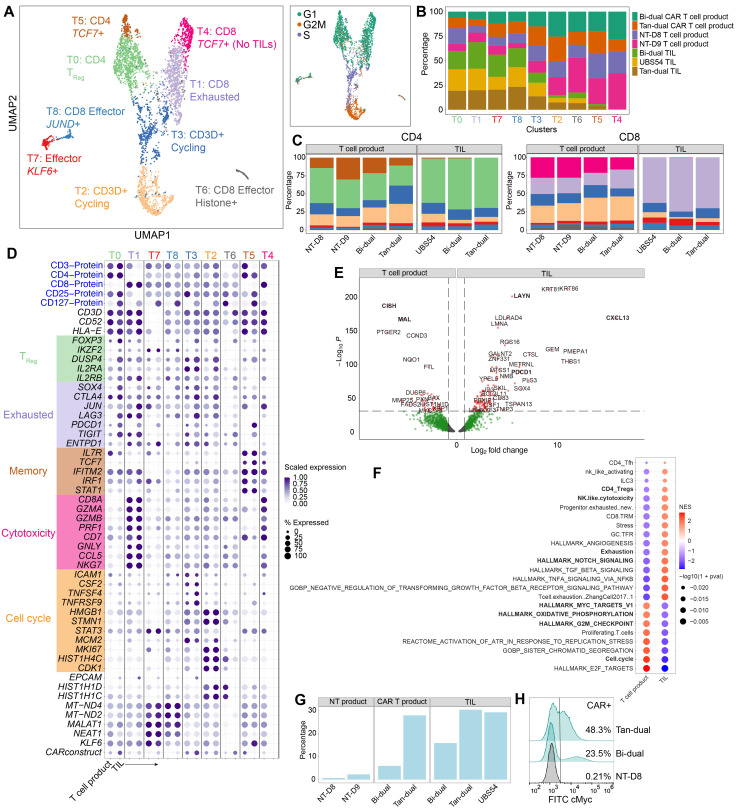
Single-cell multiomics of T cells. (**A**) UMAP based on the scRNA-seq analysis of T cells from both control (non-transduced T and CAR T cell products) and tumor-infiltrating CAR T cells. The zoomed-in inset is the same UMAP, color coded by cell cycle signature. (**B**) Stacked bar plot representing the proportion of cells in each cluster from each sample (color coded). (**C**) Stacked bar plots representing the percentage of CD4 and CD8 T cells in each cluster. Color code represents clusters. (**D**) Dot plot presenting the differentially expressed genes between clusters and split by TILs versus T cell product controls (non-transduced T and CAR T cell products). Color code represents clusters. (**E**) Volcano plot of differentially expressed genes between TILs versus T cell product controls. (**F**) GSEA of differentially expressed genes between TILs versus T cell product controls. (**G**) Proportion of cells with a detectable CAR construct in each sample. (**H**) CAR expression in NT-D8, Bi-dual, and Tan-dual T cell products detected by flow cytometry using an antibody against the cMyc tag which was introduced at the N-terminus of each CAR construct.

**Figure 4 cancers-15-05552-f004:**
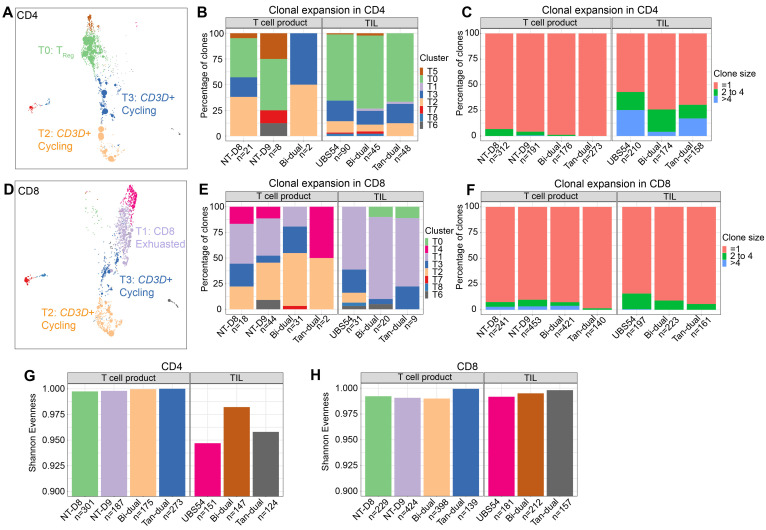
Analysis of clonal expansion using TCR sequences. (**A**) Distribution of clonally expanded CD4 cells on the UMAP. (**B**) Distribution of clones of CD4 T cells with size >1 by sample of origin (columns) and clusters (colors). The number of clones in each column is shown in the *x* axis. (**C**) Distribution of clone size in CD4 T cells by origin of sample. The number of clones in each column is shown in the *x* axis. (**D**) Distribution of clonally expanded CD8 ells on the UMAP. (**E**) Distribution of clones of CD8 T cells with size > 1 by origin of sample (columns) and clusters (colors). (**F**) Distribution of clone size in CD8 T cells by origin of sample. (**G**,**H**) Diversity (Shannon entropy) of the TCR beta chain motifs in the CD4 (**G**) and CD8 (**H**) compartment of clonally expanded TILs identified with GLYPH2 by cluster (color).

**Figure 5 cancers-15-05552-f005:**
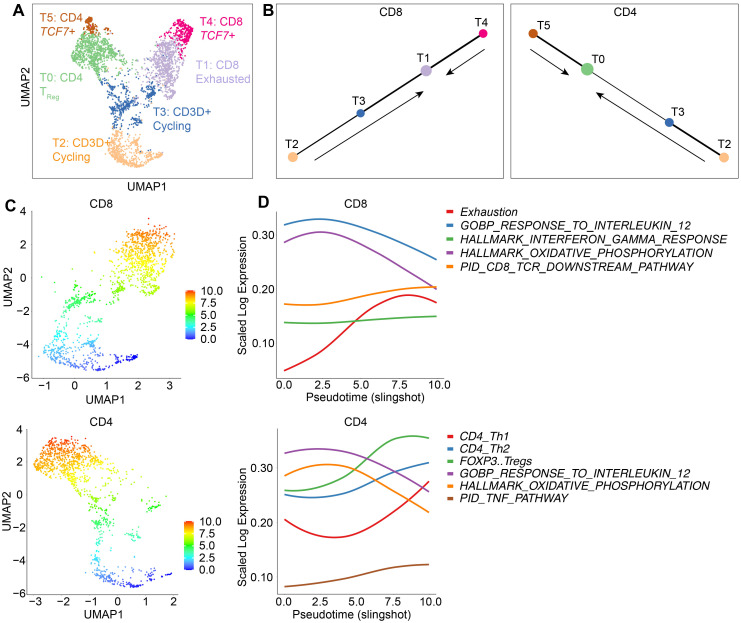
Trajectory analysis of CD4 and CD8 reveals differentiation of CAR T cells towards clonally expanded exhausted and T_Reg_ subsets during the interaction with tumor cells. (**A**) Clusters identified for trajectory analysis, excluding the small isolated clusters T6, T7, and T8. (**B**) PAGA plots showing the connectivity between clusters of CD4 and CD8 T cells, respectively. Arrows indicate the direction of the pseudotime trajectory. (**C**) The UMAP clusters displaying the pseudotime values of CD4 and CD8 T cells. (**D**) Gene signature AUC curves along the inferred pseudotime for CD4 and CD8 T cells. Fit of the gene expression along pseudotime are performed using Loess fit.

**Table 1 cancers-15-05552-t001:** Samples and conditions used for single-cell analyses.

Cell Type	Conditions or Cohorts	Number of Cells	Surface Markers	Type of Analyses
Pre-xenograft MKN-45	Pre-xenograft	3300	ICAM-1, EpCAM, HT1	GEX, Surface
Xenograft MKN-45	No T cohort	3300	ICAM-1, EpCAM, HT2	GEX, Surface
	NT (donor 9)	3300	ICAM-1, EpCAM, HT3	GEX, Surface
	UBS54 (donor 9)	3300	ICAM-1, EpCAM, HT4	GEX, Surface
	Bi-dual (donor 9)	3300	ICAM-1, EpCAM, HT5	GEX, Surface
	Tan-dual (donor 8)	3300	ICAM-1, EpCAM, HT6	GEX, Surface
T cell products	NT cell (donor 9)	2500	CD3, CD4, CD8, CD25, CD127, HT8	GEX, Surface, V(D)J
	NT cell (donor 8)	2500	CD3, CD4, CD8, CD25, CD127, HT1, HT2	GEX, Surface, V(D)J
	Bi-dual CAR T cell (donor 9)	2500	CD3, CD4, CD8, CD25, CD127, HT1, HT3	GEX, Surface, V(D)J
	Tan-dual CAR T cell (donor 8)	2500	CD3, CD4, CD8, CD25, CD127, HT1, HT4	GEX, Surface, V(D)J
TILs	UBS54 TIL (donor 9)	3300	CD3, CD4, CD8, CD25, CD127, HT1, HT5	GEX, Surface, V(D)J
	Bi-dual TIL (donor 9)	3300	CD3, CD4, CD8, CD25, CD127, HT1, HT6	GEX, Surface, V(D)J
	Tan-dual TIL (donor 8)	3300	CD3, CD4, CD8, CD25, CD127, HT1, HT7	GEX, Surface, V(D)J

## Data Availability

All data needed to evaluate the conclusions in the paper are present in the paper and/or the Appendix A. All materials used in this paper are commercially available or available from the corresponding author upon reasonable request. scRNA-seq data are publicly available at https://www.ncbi.nlm.nih.gov/bioproject/PRJNA1043405.

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
