# Peer review of "Chimeric Antigen Receptor T Cell Therapy Targeting Epithelial Cell Adhesion Molecule in Gastric Cancer: Mechanisms of Tumor Resistance"

_cancers, 2023, doi:10.3390/cancers15235552_

Round 1

Reviewer 1 Report

Comments and Suggestions for Authors

Yanping Yang and co-authors present a high quality and well-written experimental manuscript focused on chimeric antigen receptor T cell therapy targeting EpCAM in gastric cancer with regards to mechanisms of tumor resistance.

Authors present a research that highlights an approach that combines imaging and multiomics methodologies to concurrently characterize the evolution of tumors and the differentiation of CAR T cells.

Authors aimed to elucidate the complex interactions between tumor cells and CAR T cells targeting EpCAM in a xenograft model of gastric cancer. Using whole-body CAR T cell imaging and  single cell multiomics analysis, they noticed that within resistant tumors, CAR T cells exhibited a tendency to proliferate, but they were largely dysfunctional, losing their ability to fight cancer ef fectively. Specifically, most CD8 T cells became exhausted within tumors, while CD4 T cells transformed into regulatory T cells that can dampen the immune response. Additionally, the resistant tumor cells had specific gene changes that could promote cancer growth and make the disease more  challenging to cure. Altogether, this research provides valuable information for understanding how tumors resist CAR T-cell therapy and may guide future developments in cancer treatment.

Authors have developed CAR T cells specifically targeting EpCAM for the treatment of gastric cancer. They seeked to unravel the precise mechanisms by which tumors evade immune surveillance and develop resistance to CAR T-cell therapy. Through a combination of whole-body CAR T-cell imaging and single-cell multiomics analyses, they have uncovered intricate interactions between tumors and tumor-infiltrating lymphocytes. In a gastric cancer model, tumor-infiltrating CD8 T cells exhibited both cytotoxic and exhausted phenotypes, while CD4 T cells were mainly regulatory T cells. T cell receptor clonal analysis provided evidence of CAR T cell proliferation and clonal expansion within resistant tumors, which is substantiated by whole-body CAR T-cell imaging. Furthermore, single cell transcriptomics showed that tumor cells in mice with refractory or relapsing outcomes were enriched for major histocompatibility complex and antigen presentation pathways, interferon-? and interferon-? responses, mitochondrial activities, and a set of genes (e.g., CD74, IDO1, IFI27) linked to tumor progression and unfavorable disease prognosis. 

Authors suggest that:
- CAR T cell imaging reveals penetration and expansion of CAR T cells in tumor
- Single cell analysis of tumor cells revealed a remodeling towards external stimuli following CAR T cell infusion
- T cell analysis revealed a rapid differentiation process that resulted in CD8 T cell exhaustion and CD4 T cells adopting a regulatory phenotype
- Analysis of TCR sequences revealed clonally expanded CAR T cells in the tumor
- Trajectory analysis demonstrates a differentiation of CAR T cells towards exhausted and TReg cells

Finally, authors conclude that their research introduces a tool to investigate the interactions between cancer and CAR T cells, and highlights the feasibility of integrating genomics and imaging techniques to investigate evolutionary process of both CAR T and cancer cells. It allows for the quantification of tumor size, identification of phenotypic and transcriptional features of tumor and T cells, as well as T cell clonal repertoire, enabling a comprehensive understanding of the molecular mechanisms driving a successful CAR T cell response.

Overall, the manuscript is highly valuable for the scientific community and should be accepted for publication.

======================

Other comments to authors:

1) Please check for typos throughout the manuscript.

2) Please improve figures/tables where appropriate.

3) With regards to using CAR-T cell therapy against solid tumors – authors are kindly encouraged to cite the following article that describes the development of solid tumor models for evaluating CAR-T cell efficacy.
DOI: 10.3390/biomedicines11020626

Author Response

1) Please check for typos throughout the manuscript.

Thank you for your feedback. We’ve carefully reviewed the manuscript to ensure that it is error-free.

2) Please improve figures/tables where appropriate.

We’ve worked on improving the quality and clarity of the figures and tables.

3) With regards to using CAR-T cell therapy against solid tumors – authors are kindly encouraged to cite the following article that describes the development of solid tumor models for evaluating CAR-T cell efficacy. 
DOI: 10.3390/biomedicines11020626

We have added the citation of this paper in the Discussion section.

Reviewer 2 Report

Comments and Suggestions for Authors

The manuscript “Chimeric antigen receptor T cell therapy targeting EpCAM in gastric cancer: mechanisms of tumor resistance” by Yanping Yang and colleagues deals with highly intricate process of interaction between tumor cells and TIL/CAR-T cells. Authors use the xenotransplantation model of human gastric cancer which is subjected to treatment with human CAR-T cells. Tumors established in NSG mice change their growth following introduction of CAR-T cells directed at EpCAM alone, or combination of EpCAM and ICAM-1. The latter are administered as either separate CAR molecules with different antigen-binding domains, or single CAR molecule in which both antigen-binding domains are presented. The aim of the study is to track changes in tumor growth and phenotypic changes in tumor cells happening in the process of immune response by TILs (in NT CAR-untreated mice) and compare with changes in response to CAR-T therapy. On the other side, the changes in T cells (TILs and CAR-T cells) are also tracked using imaging technics, and, more importantly, is supported by analysis of gene expression profiles and surface phenotype of T cells. Not very surprisingly, exhaustion of T cells (TILs and CAR-T cells) is observed in recipients that demonstrate tumor growth. Single cell analysis shows that exhaustion marks in T cells grossly correlate with parallel changes in tumor cells. Trajectories of CD8 and CD4 T cells demonstrate that CD8 T cells differentiate to CTLs or exhausted cells, while CD4 T cells similarly become effector cells, exhausted cells and, in addition, suppressive Treg cells.   

Without any concern, the study is very solid. It is well-designed, experimentally perfect and is of interest to those who study tumor microenvironment and anti-tumor immunity. The data are new, their interpretation is mostly clear. The manuscript is well written and illustrated. The scope of the study is of great value, but according to authors, their work has significant limitations. It should be noted, that these limitations and several minor points should be better cleared prior to acceptance of the manuscript to Cancers.

Below is the list of concerns that should be addressed:

1)     In discussion, the authors mention that their study relies on analysis of tumor cells, TILs and CAR-T cells isolated from only one animal. Understandably, lack of statistic in this case is compensated by single-cell sequencing analysis of isolated populations of cells. Nevertheless, it would be great to show that in animals responding to therapy and showing reduction in tumor size, exhaustion of T cells is less profound, and proportion of exhausted cells is lower, compared to mice that fail to respond. Reference to negative control is not helpful since introduction of CAR-T cells should affect not only tumor cells but also antigen presenting cells and other components of tumor stroma;

2)     It has to be stressed that instead of mentioning that single mouse and single tumor per each group was taken for analysis in the very end, it should be placed to Material and methods section, to Figure legends and section 3.2 of the Results. Otherwise it is not obvious what was really done!

3)     In Figure 3G, authors show expression of transcripts that code for CAR proteins. It is very surprising to see that about 2% of non-transduced cells display the presence of this transcript. This should be explained carefully, since 2% is relatively high and is comparable to percentage of cells placed to some clusters from Fig. 3C. It is also true for analysis of clonality of TILs and CAR-T cells presented in Fig.4.

4)     Figure 3G, typo - instead of NT-D9 one can see DT-D9.

As it was mentioned earlier, the system, authors try to study, is not bimodal. Aside from T cells and tumor parenchymal cells, there are multiple different cell types and induced populations that should affect the outcome of immune response to tumor. It would be great if a couple of sentences would be added to the main text to avoid oversimplification of tumor – T cell relations and explain that at least some of interactions are indirect and include other cell types. 

Author Response

1)     In discussion, the authors mention that their study relies on analysis of tumor cells, TILs and CAR-T cells isolated from only one animal. Understandably, lack of statistic in this case is compensated by single-cell sequencing analysis of isolated populations of cells. Nevertheless, it would be great to show that in animals responding to therapy and showing reduction in tumor size, exhaustion of T cells is less profound, and proportion of exhausted cells is lower, compared to mice that fail to respond. Reference to negative control is not helpful since introduction of CAR-T cells should affect not only tumor cells but also antigen presenting cells and other components of tumor stroma.

We have added further clarification in Discussion regarding the limitations of the present study:

“However, we also acknowledge several limitations of this study. Firstly, single cell studies were conducted using tumor cells and TILs isolated from mice with sufficient tumor mass and TIL infiltration. Tumor-infiltrating CAR T cells in our study exhibited a phenotypic spectrum ranging from cytotoxic T cells at one end to exhausted T cells at the other, reflecting the partial suppression of tumor growth by CAR T cells. Our analysis did not include CAR T cells from mice with complete tumor response, primarily due to technical challenges associated with collecting these T cells. After tumor elimination, CAR T cells often contract or migrate away from the tumor sites. Nevertheless, it is reasonable to speculate that CAR T cells from complete responders would exhibit a phenotypic spectrum skewed towards the cytotoxic end, featuring fewer exhausted and dysfunctional T cells.”

2)     It has to be stressed that instead of mentioning that single mouse and single tumor per each group was taken for analysis in the very end, it should be placed to Material and methods section, to Figure legends and section 3.2 of the Results. Otherwise it is not obvious what was really done!

Thank you for the suggestion. We have now included this information in the Material and Methods section and Figure legends.

3)     In Figure 3G, authors show expression of transcripts that code for CAR proteins. It is very surprising to see that about 2% of non-transduced cells display the presence of this transcript. This should be explained carefully, since 2% is relatively high and is comparable to percentage of cells placed to some clusters from Fig. 3C. It is also true for analysis of clonality of TILs and CAR-T cells presented in Fig.4.

We pooled multiple hash-tagged T cell samples before generating the GEMs. The presence of the CAR transcripts in non-transduced T cell samples is likely the result of imprecise demultiplexing of the hash-tagged NT and CAR T cell samples. We added explanation in the Results section 3.4.

4)     Figure 3G, typo - instead of NT-D9 one can see DT-D9.

We have corrected the typo.

As it was mentioned earlier, the system, authors try to study, is not bimodal. Aside from T cells and tumor parenchymal cells, there are multiple different cell types and induced populations that should affect the outcome of immune response to tumor. It would be great if a couple of sentences would be added to the main text to avoid oversimplification of tumor – T cell relations and explain that at least some of interactions are indirect and include other cell types. 

We have discussed this limitation in the discussion section:

“Our study used an immunocompromised NSG model, and as a result, it did not fully recapitulate the complexity of the tumor stroma and the interactions among tumor stroma comprising T cells and other immune cells. Despite this limitation, our approach allowed us to focus on the investigation of cellular interactions between tumor cells and TILs without being complicated by other cellular components. Moreover, NSG tumor models have been widely utilized in the preclinical evaluation of CAR T cell therapy, offering greater clinical relevance compared to in vitro models.”

Reviewer 3 Report

Comments and Suggestions for Authors

The article entitled ‘Chimeric antigen receptor T cell therapy targeting EpCAM in gastric cancer: mechanisms of tumor resistance’ was well received. The study seems interesting and carefully designed. The introduction section provides sufficient background for the research. The results are clearly presented and the conclusions drawn are supported by results. The discussion section contains the limitations of current study which is appreciable.

However discussion section provides little supporting literature and it is advised to add some more references into the discussion section.  

Please state the reason behind CAR-T cell exhaustion in tumor microenvironment with respect to the current study.

Comments on the Quality of English Language

minor spell check required

Author Response

However discussion section provides little supporting literature and it is advised to add some more references into the discussion section.  

Please state the reason behind CAR-T cell exhaustion in tumor microenvironment with respect to the current study.

We appreciate the reviewer’s suggestion. We have added more literatures on CAR T-tumor interaction and discussed the reason behind the CAR T cell exhaustion in tumor microenvironment.

“The presence of dysfunctional yet proliferating T cells aligns with recent observations in TILs isolated from melanoma and NSCLC cells [4; 5]. A study of CAR T cells targeting mesothelin in preclinical mouse models also identified proliferating and hypofunctional CAR T cells [7]. These tumor-infiltrating CAR T cells  underwent rapid loss of functional activity, characterized by upregulation of intrinsic T cell inhibitory enzymes and surface inhibitory receptors (PD1, LAG3, TIM3 and 2B4) [7]. By integrating single cell multiomics and spatiotemporal CAR T cell imaging techniques, our study contributes further insights into the interaction between CAR T cells and tumors.”

“In our immunocompromised NSG model, CAR T exhaustion is likely driven primarily by chronic interactions between CAR T cells and persistent  tumors. When antigens persist, such as in chronic infections and resistant cancers, T cells can enter a dysfunctional state [46]. Tonic signaling mediated by CARs can also accelerate T cell differentiation and exhaustion [47]. The design of the CAR construct, the promoter, and viral vector used to express the CAR should all impact on the level of tonic signaling, thereby influencing T cell exhaustion [48; 49]. Another contributing factor could be the inadequate vasculature within the rapidly growing xenograft tumors, leading to hypoxic conditions and contributing to the generation of exhausted CD8 T cells and TReg cells [50; 51].”

Reviewer 4 Report

Comments and Suggestions for Authors

CAR T cell-based therapy revolutionized. treatment paradigm in lymphatic hematological malignancies but are in their infancy as for solid tumors beside some promising results in tumors. Gliomas and some subtypes of medulloblastoma. Understanding solid tumor resistance to CAR T cell therapy is therefore an unmet need.  Yanping Yang and colleagues address this issue in a very methodical way using some of the most updated imaging and single cell techniques including whole-body CAR T-cell imaging and single-cell multiomics analyses. The authors  developed 3 types of CAR T cells against  anti EpCAM and ICAM (single ,Bi-dual or Tan-dual CARs and assessed their function in animal gastric carcinoma .They revealed   the presence of dysfunctional, yet proliferating  CAR –T cells that were not described before for CAR-T cells   and dissected the complex interactions between CAR T and tumor cells in the tumor microenvironment  involved molecular mechanisms associated with both T cell activation and suppression trying to elucidate mechanisms of resistance. I congratulate the authors for a cutting-edge study and have no specific comments.

Author Response

We thank the reviewer for dedicating time and effort to review our manuscript.